# Pediatric Glioma: An Update of Diagnosis, Biology, and Treatment

**DOI:** 10.3390/cancers13040758

**Published:** 2021-02-12

**Authors:** Yusuke Funakoshi, Nobuhiro Hata, Daisuke Kuga, Ryusuke Hatae, Yuhei Sangatsuda, Yutaka Fujioka, Kosuke Takigawa, Masahiro Mizoguchi

**Affiliations:** Department of Neurosurgery, Graduate School of Medical Sciences, Kyushu University, 3-1-1 Maidashi, Higashi-Ku, Fukuoka 812-8582, Japan; sf1wan0610@gmail.com (Y.F.); kuga@ns.med.kyushu-u.ac.jp (D.K.); ryhatae@ns.med.kyushu-u.ac.jp (R.H.); y-sangat@med.kyushu-u.ac.jp (Y.S.); yfujioka@med.kyushu-u.ac.jp (Y.F.); taki1221@med.kyushu-u.ac.jp (K.T.); mmizoguc@ns.med.kyushu-u.ac.jp (M.M.)

**Keywords:** pediatric glioma, molecular profiling, next-generation sequencing, molecular targeted therapy, cIMPACT-NOW

## Abstract

**Simple Summary:**

Recent research has enhanced our understanding of the diverse biological processes that occur in pediatric gliomas; and molecular genetic analysis has become essential to diagnose and treat these conditions. Because targetable molecular aberrations can be detected in pediatric gliomas, identifying these aberrations is very important. This review provides an overview of pediatric gliomas, and describes recent developments made in strategies for their diagnosis and treatment. Additionally, it presents a current picture of pediatric gliomas in light of advances in molecular genetics, and describes the current scientific progress in gliomas’ treatment using information from recently completed and ongoing clinical trials. The era of incorporating molecular genetic analysis into clinical practice is emerging.

**Abstract:**

Recent research has promoted elucidation of the diverse biological processes that occur in pediatric central nervous system (CNS) tumors. Molecular genetic analysis is essential not only for proper classification, but also for monitoring biological behavior and clinical management of tumors. Ever since the 2016 World Health Organization classification of CNS tumors, molecular profiling has become an indispensable step in the diagnosis, prediction of prognosis, and treatment of pediatric as well as adult CNS tumors. These molecular data are changing diagnosis, leading to new guidelines, and offering novel molecular targeted therapies. The Consortium to Inform Molecular and Practical Approaches to CNS Tumor Taxonomy (cIMPACT-NOW) makes practical recommendations using recent advances in CNS tumor classification, particularly in molecular discernment of these neoplasms as morphology-based classification of tumors is being replaced by molecular-based classification. In this article, we summarize recent knowledge to provide an overview of pediatric gliomas, which are major pediatric CNS tumors, and describe recent developments in strategies employed for their diagnosis and treatment.

## 1. Introduction

Central nervous system (CNS) tumors are the most frequent solid tumors in children, accounting for 21% of pediatric cancers and representing a primary cause of mortality [1]. Survivors of pediatric CNS tumors are at a high risk for late mortality and for developing subsequent neoplasms and chronic diseases [2]. The main secondary disorders are endocrine and cognitive disorders. In addition, only half of the patients are normally employed when they become adults, due to their complex disabilities [3]. Improvement of diagnosis and treatment of pediatric CNS tumors, and care of child-to-adult transition for these patients, are required [3].

Recent research has promoted our understanding of the complex biology of pediatric CNS tumors. Molecular genetic analysis is essential not only for proper classification but also for monitoring biological behavior and clinical management of tumors. Recent genome studies have revealed several unique genomic changes observed in pediatric CNS tumors, which were different from those observed in adults [4,5,6]. Since the 2016 World Health Organization (WHO) classification of CNS tumors, molecular profiling has become an indispensable step in the diagnosis, prognosis, and treatment of pediatric as well as adult tumors of this type [7]. These molecular data are changing the way tumors are being diagnosed. Specifically, morphology-based classification of tumors is being replaced by molecular-based classification, leading to new guidelines and offering novel molecular targeted therapies. The Consortium to Inform Molecular and Practical Approaches to CNS Tumor Taxonomy (cIMPACT-NOW) is making practical recommendations based on recent advances in CNS tumor classifications, particularly those based on molecular discernment of these neoplasms [8]. In this article, we summarize our current understanding of the most common pediatric gliomas, which are the major types of pediatric CNS tumors, and describe recent developments in strategies of their diagnosis and treatment.

## 2. Diagnostic Approach for Pediatric Brain Tumors

### 2.1. Integrated Diagnosis with Histological and Genetical Classification

In the past, conventional diagnosis of pediatric CNS tumors involved a pathologist’s histopathologic review, supported by radiological findings, such as magnetic resonance imaging, and standardized immunohistochemical testing for specific biomarkers. However, the importance of molecular genetic analysis has become a focus over the past decade, and the molecular profiling of tumors has revealed a number of important details including prognostic factors and predictive markers of drug sensitivity, resistance, and adverse events. This has allowed personalized therapy to become feasible. Molecular genetic analysis is required in order for pediatric CNS tumors to be incorporated into a histologically based tiered classification system and improve their diagnosis. In the 2016 CNS WHO classification of medulloblastomas, a histologically defined and a genetically defined lists of tumors were combined to produce an integrated list of parameters for improved diagnosis [9]. This provided great flexibility, while at the same time conveyed key diagnostic information in a layered concise format [10]. In cIMPACT-NOW update 6, the clinical utility of this combined two-list approach, has been proven to optimally categorize pediatric-type glial/glioneuronal tumors and ependymomas [8].

### 2.2. DNA Methylation Profiling

DNA methylation-based classification for CNS tumors is reported to be a reproducible and valuable approach, and is expected to reduce the substantial inter-observer variability that occurs in conventional morphology-based classification [11]. In cIMPACT-NOW update 6, although the committee did not recommend methylation profiling as the only method to identify specific tumor types or subtypes, they agreed that many CNS tumor types and subtypes can be reliably identified using this approach [8]. Methylation profiling for pediatric CNS tumors is a useful tool to identify specific subtypes with different clinical outcomes, and is expected to be a robust tool for diagnosis [11,12]. In pediatric CNS tumors, methylation profiling provides a reproducible modality for classification with high concordance to subtypes initially identified by gene expression profiling and genome sequencing [12].

### 2.3. Molecular Approaches for Precision Therapy

In recent years, advances in next-generation sequencing (NGS) and array-based genomic platforms have transformed analysis of the molecular landscapes of cancers, including pediatric CNS tumors [13]. NGS-based profiling allows analysis of some combinations of germline and/or somatic single nucleotide variants, small insertions, or deletions, and structure variants in the form of DNA copy number alterations, translocations, inversions, and other more complex alterations [13]. This novel method makes genomic-based precision therapy feasible. A national trial of pediatric cancer using molecular analysis for treatment decisions is currently being conducted by the Children’s Oncology Group [14]. Known as the National Cancer Institute-Molecular Analysis for Therapy Choice (NCI-MATCH) trial, eligibility for assignment to treatment arms is determined based on predefined lists of genomic aberrations. In this trial, the tissue of solid tumors from pediatric and adolescent patients have undergone molecular profiling. If an aberration that has been defined as a driver mutation for a MATCH study drug targeting the identified aberration is identified, the patient has the opportunity to enroll in the relevant single agent treatment arm [14].

Liquid biopsy is also one of the current topics as a less invasive method using body fluid, such as plasma or cerebrospinal fluid (CSF), for molecular diagnosis. Recent efforts including ours have developed digital PCR-based liquid biopsy targeting cell-free tumor DNA in CSF for detecting glioma-specific diagnostic mutations [15]. NGS-based liquid biopsy has been attempted. Miller et al. used NGS to analyze CSF samples from patients with diffuse gliomas, and identified glioma-related genetic alterations in 42 (49.2%) of 85 tumors [16]. Although they also detected these alterations from 3 (15.8%) of 19 plasma samples, all patients with positive plasma had radiological evidence for dissemination within the CNS [16]. These results indicated that the sensitivity of liquid biopsy is still an unsolved issue for the molecular diagnosis of CNS tumors. Accordingly, sufficient tissue samples obtained by conventional biopsy seems to be essential in the present state, especially for NGS or methylation profiling.

Because targetable molecular aberrations can be detected in pediatric CNS tumors, searching for these aberrations is essential for treating children with these tumors. However, most existing cancer multi-gene panel tests using NGS are currently performed only for adult cancer, resulting in undetectability of the rare genetic aberrations specific to pediatric cancer. Although molecular diagnosis is essential in cases of pediatric cancer, current cancer multi-gene panel tests cannot be used for the purpose of diagnosis. To resolve this issue, Kohsaka et al. reported the establishment of a comprehensive assay, the Todai OncoPanel, which consists of DNA and RNA hybridization capture-based NGS panels [17]. In this method, fusion genes, which are frequently detected in pediatric CNS tumors, can be accurately and cost-effectively identified because of the development of the junction capture method for RNA sequencing [17]. The development and implementation of NGS panels for pediatric cancer diagnosis is coming of age.

Currently, novel molecular profiling technologies, including whole-proteome, phosphoproteome, metabolome, and single-cell RNA-sequencing are also developing. Many of these technologies are being applied in current research to further characterize tumor biology [13]. In addition to development of the molecular-based diagnostic technologies, international cooperation to identify and approve new pediatric oncology drugs, such as ACCELERATE organized in Europe in 2015, has been advancing [18]. Many clinical trials for novel treatment are being conducted around the world. The era of incorporating molecular genetic analysis into clinical practice is beginning (Figure 1).

## 3. Histopathologic Subtypes of Pediatric CNS Tumors

Numerous entities comprise pediatric CNS tumors. The Central Brain Tumor Registry of the United States (CBTRUS) Statistical Report showed the frequency of pediatric CNS tumors in a population-based study in the USA [19]. Gliomas were the most common. The majority of pediatric gliomas are pediatric low-grade glioma (pLGG) classified as WHO grade 1 or 2, but some develop in a short time period and progress rapidly, classified into WHO grade 3 or 4 as pediatric high-grade glioma (pHGG) [20]. According to the report, pilocytic astrocytoma (PA) accounted for 17.5%, other astrocytomas accounted for 8.9%, and mixed glioneuronal tumors accounted for 6.5% of pediatric CNS tumors. Oligodendroglial tumors were rare, with oligodendrogliomas at 0.9%, of cases and oligoastrocytic tumors at 0.6%. Glioblastoma (GBM) was also rare, accounting for 2.6% of cases. HGGs not otherwise specified (NOS) were 14.2% of cases. Ependymal tumors were 5.6%, and germ cell tumors were 3.7% of cases. As benign tumors, nerve sheath tumors, craniopharyngioma, and pituitary tumors comprised 4.8%, 4.0%, and 3.8% of cases, respectively. Others made up 9.3% of cases. Because these tumors are fundamentally different from those occurring in adults, specific treatment for (and management of) younger individuals are important.

## 4. Low Grade Gliomas

Compared to adult LGGs, isocitrate dehydrogenase (IDH) mutations are less observed in children, and malignant progression is extremely rare. These gliomas account for approximately 30% of pediatric CNS tumors [21,22]. The 10-year overall survival (OS) is high, over 95%, but 10-year progression-free survival (PFS) is only approximately 50%, and half of patients require adjuvant therapy [23]. Surgical resection is important for the management of pLGG, and complete resection is the most favorable predictor of survival in patients with pLGG [24]. In patients where gross tumor removal cannot be achieved, progression of the tumor has been treated with adjuvant chemotherapy and radiation. However, patients with unresectable tumors have chronic clinical conditions and experience long-term reduction in quality of life [25]. In particular, radiation is associated with increased mortality [26,27,28].

### 4.1. Molecular Landscape in pLGG

#### 4.1.1. RAS/Mitogen-Activated Protein Kinase (MAPK) Pathway

In pLGG, the most common entity is pilocytic astrocytoma (PA) comprising >15% of tumors in patients aged 0 to 19 years [29]. PA is classified into WHO grade 1, and has a 10-year survival of over 90%, although a rare variant termed “pilomyxoid astrocytoma”, which occurs predominantly in children under 1 year of age, and in the hypothalamic/chiasmatic region, has been classified as WHO grade 2. As a molecular aberration of PA, *KIAA1549*-*BRAF* fusion by tandem duplication is historically well-known and the most frequently observed (>70%) [30]. However, high-throughput sequencing techniques that interrogate the whole genome have shown that a single aberration of the RAS/MAPK pathway is exclusively found in almost all cases, indicating that PA represents a one-pathway disease [30]. Recent studies have put forth an overview of the RAS/MAPK pathway alterations in pLGG [30,31,32,33]. Typical aberrations are reported as follows. 

*BRAF* V600E mutation: in pLGG, patients with the *BRAF* V600E mutation demonstrated poor outcomes. Lassaletta et al. reported that the 10-year PFS is 27% and 60.2% for the *BRAF* V600E mutant and wild-type pLGG, respectively (*p* < 0.001) [34]. *BRAF* V600E mutations can be found in CNS tumors in any location, and are often detected in midline tumors, including the optic pathway, brainstem, and spinal cord [34]. The *BRAF* V600E mutation is commonly observed in pleomorphic xanthoastrocytomas (78%), followed by gangliogliomas (49%), and also in diffuse astrocytomas (43%) [34]. Astroblastomas are also known to harbor *BRAF* V600E [35]. However, this mutation is rare in PAs (3%) [34].

*FGFR1*: *FGFR*s are a family of receptor tyrosine kinases [36]. Their dysregulation has been detected in a wide variety of cancers, such as urothelial carcinoma, hepatocellular carcinoma, ovarian cancer, and lung adenocarcinoma [37]. *FGFR1-3* is identified in CNS tumors, including glioma, ependymoma, and medulloblastoma [37,38,39]. In pediatric gliomas, *FGFR1* is well-noted and is expected to be a targetable aberration. *FGFR1* aberrations include *FGFR1* mutations, *FGFR1-TACC1* fusions, and *FGFR1-TKD* duplications [33,40,41]. These aberrations cause *FGFR1* autophosphorylation, resulting in upregulation of the RAS/MAPK pathway (Table 1) [33,40].

*NF-1*: neurofibromatosis type 1 (NF-1) is a well-known inherited tumor predisposition syndrome caused by a germline mutation in the *NF-1* tumor suppressor gene. This gene encodes neurofibromin, a GTPase-activating protein that functions as the negative regulator of RAS [42,43]. LGGs in the optic pathway are diagnosed in 10–15% of children with NF-1 [44,45]. Most LGGs associated with NF-1 are benign and require no treatment. However, LGGs associated with NF-1 in younger children under 2 years of age and/or outside of the optic pathway have risk of progression and poor outcomes [23,46].

#### 4.1.2. Non-RAS/MAPK Pathway

pLGG, including PA, is considered a single RAS/MAPK pathway disease. However, some aberrations indirectly affect the RAS/MAPK pathway.

*MYB* and *MYBL1* alterations: *MYB* alteration influences on control of proliferation and differentiation of hematopoietic and other progenitor cells, and is associated with proto-oncogenic functions in both human leukemia and solid tumors [47,48]. Bandopadhayay et al. reported that 10% of pLGGs contained *MYB* alterations, and *MYB-QKI* fusion was specified as a driver aberration for angiocentric gliomas [49]. *MYBL1* is a member of the *MYB* family. Although the biological function of *MYBL1* is less known than that of *MYB*, *MYBL1* is considered to function as a transcriptional regular critical for proliferation and differentiation [23]. *MYBL1* alterations are rare and are detected in diffuse astrocytomas [41,49]. pLGGs frequently have alterations in *MYB* family genes, such as *MYB* and *MYBL1* [33,50]. These alterations are detected more frequently in young children (median age, 5 years) and often occur in the cerebral hemispheres [51]. The clinical course is generally indolent in glioma patients with *MYB* and *MYBL1* alterations, and Chiang et al. reported that the 10-year OS is 90%, and 10-year PFS is 95% [51]. 

*CDKN2A* homozygous deletion: *CDKN2A* is a gene encoding two tumor suppressors, protein p14^ARF^, and p16^INK4A^. Homozygous deletion of *CDKN2A* can contribute to uncontrolled tumor cell proliferation [52], and has been reported as a poor prognostic marker in adult glioma [53,54,55]. *CDKN2A* homozygous deletion is also observed in pediatric gliomas, and is a well-known hallmark lesion of pleomorphic xanthoastrocytoma. *CDKN2A* homozygous deletion co-occurs with the *BRAF* V600E mutation, demonstrating poor clinical outcomes [7,56].

### 4.2. Integrated Diagnosis of Pediatric Diffuse Gliomas in cIMPACT-NOW Update 4

In cIMPACT-NOW update 4, an integrated diagnosis of pediatric diffuse gliomas was reported. In this report, pediatric diffuse gliomas were classified by *MYB*, *MYBL1*, or *FGFR1* alterations or *BRAF* V600E mutations [57]. These aberrations are not frequent; however, Qaddoumi et al. reported that a *BRAF* V600E mutation, *FGFR* alteration, or rearrangement of *MYB* or *MYBL1* were detected in 84% of *IDH*-wild type/*H3*-wild type diffuse gliomas in a large pediatric cohort [41]. The following classifications were considered to provide valuable diagnostic and prognostic information, and for some entities, suggest targeted therapies [57]:Diffuse glioma, *MYB*-altered;Diffuse glioma, *MYBL1*-altered;Diffuse glioma, *FGFR1 TKD*-duplicated;Diffuse glioma, *FGFR1*-mutant;Diffuse glioma, *BRAF* V600E-mutant;Diffuse glioma, other MAPK pathway alteration.

### 4.3. Targeted Therapy for pLGG

We summarize recently reported clinical trials of targeted therapy for pediatric glioma in Table 2 and ongoing clinical trials in Table 3. Successful treatment with *BRAF* inhibitors for *BRAF*-mutated gliomas has been recently reported [58,59,60,61]. Dabrafenib, one of those agents, is expected to improve clinical outcomes with few adverse events and good tolerance in patients with *BRAF*-mutated gliomas [60,61,62]. Despite high initial response rates, acquired resistance to *BRAF* inhibitors occurs in a majority of patients, and one of the most frequent causes of this resistance is reactivation of the MAPK pathway [63]. However, an additional *MEK* inhibitor that inhibits the MAPK pathway to *BRAF* inhibitors, trametinib, overcame *BRAF* inhibitor resistance, and demonstrated superiority over a *BRAF* inhibitor alone in phase III clinical trials in patients with *BRAF* mutant metastatic melanoma [64,65]. In CNS tumors, including gliomas, clinical experience with a combination of *BRAF* and *MEK* inhibitors has also been reported [61,66,67]. To evaluate the efficacy and safety of dabrafenib in combination with trametinib for pediatric gliomas, a nationwide phase II pediatric study is in progress. In a recent phase II study of patients with pLGG in the United States, the efficacy of selumetinib, a *MEK1/2* inhibitor, was assessed [68]. In this study, children with PA harboring either one of the two most common *BRAF* aberrations, *KIAA1549*-*BRAF* fusion or the *BRAF* V600E mutation, and NF-1 associated LGG, participated. Selumetinib was found to be active in cases of recurrent or progressive *BRAF* aberrated PA and NF-1 associated LGG [68]. In Japan, a selumetinib pediatric NF-1 phase I study is in progress, although this study does not include CNS tumors. Early clinical approval and expanded indication for selumetinib in treating CNS tumors is expected. In addition, in a phase I study, vemurafenib, a *BRAF* V600E inhibitor approved for metastatic melanoma, was determined to be acceptable for recurrent or refractory *BRAF* V600E mutant glioma [69,70]. In the near future, these targeted therapies may be included among standard chemotherapies for treatment of *BRAF*-mutated tumors.

*FGFR* kinase inhibitors are currently in clinical development. AZD4547 is an orally bioavailable *FGFR1-3* inhibitor [71]. In the NCI-MATCH trial, AZD4547 was administered to patients with tumors harboring *FGFR1-3* mutations or fusions. In this phase II trial, the CNS tumor was only one case out of a total of 20, and AZD4547 was not found to meet the primary end point. However, this trial showed modest activity of AZD4547 in patients with *FGFR* mutations and fusions, suggesting the possibility of clinical use in the future [72].

## 5. High Grade Gliomas

pHGGs account for 8–12% of pediatric CNS tumors, including anaplastic astrocytoma (WHO grade 3) and GBM (WHO grade 4), both types malignant, diffuse, infiltrating astrocytic tumors [7,77,78]. Current management of pHGG consists of maximal surgical resection followed by radiotherapy with concurrent and adjuvant alkylator therapy. These gliomas have a 3-year PFS of 10% and 3-year OS of 20% [79]. Diffuse intrinsic pontine glioma (DIPG), which is an exceptionally unresectable tumor, has an aggressive clinical course, even when the histological findings demonstrate low-grade [80]. This feature is reflected in the 2016 WHO classification—diffuse midline gliomas with *K27M* histone mutations, including most DIPGs, are classified as WHO grade 4, regardless of histological findings [7,81].

### 5.1. Molecular Landscape in pHGG (Table 1)

*G34R/V* and *K27M* mutations are well-known mutations in pHGG. In an integrated molecular meta-analysis of pHGG, the distribution of recurrently mutated genes was 47% *TP53*, 36% *H3.3*, 24% *ATRX*, and 7% *BRAF* V600E [82]. These tumors included presence of mutually exclusive *G34* and *K27M* mutations, and co-occurring mutations of *ATRX* with *G34R* of *NF1* [83,84].

*H3K27M* mutation: histone H3 (*H3F3A* and *HIST1H3B*) *pK27M* mutations are frequently observed in DIPGs, which arise in the brainstem almost exclusively in children, and in pHGGs in midline structures such as the thalamus and spinal cord [85,86,87,88,89]. This tumor type was classified as a separate entity, “diffuse midline glioma, *H3K27M*-mutant” (WHO grade 4), in the 2016 WHO classification [7]. *H3K27M* mutation results in a global H3K27me3 reduction by multiple mechanisms, such as aberrant polycomb repressive complex 2 (PRC2) interactions and hampered H3K27me3 spreading. It is also thought to suppress neuroglial differentiation through deregulation of epigenetic modifications [90,91,92,93]. Although additional mechanisms have not been revealed to date, the *H3K27M* mutation is an important key for pHGG treatment.

*H3G34R/V* mutation: in cIMPACT-NOW update 6, diffuse glioma, *H3.3 G34*-mutant was reported as a novel tumor type separated from the established gliomas as well as from diffuse midline glioma, *H3K27M*-mutant. It was also found to correspond with WHO grade 4 [8]. When compared with the *K27M* mutation, how the *G34* mutation affects the epigenome remains unclear. *G34R/V* in *H3F3A*, which encodes non-canonical histone *H3.3*, occurs in pHGGs of the cerebral cortex [85,86,88], whereas the *G34W* mutation is prevalent in bone tumors [94]. Neumann et al. analyzed data from 77 patients with GBM under the age of 30, and found that the frequency of *G34R/V* and *K27M* mutations was 16% and 32%, respectively [95]. Among our 411 consecutive glioma patients, 14 (3.4%) harbored *H3F3A* mutations, of which 4 had *G34R* mutations and 10 *K27M* mutations [96]. We recently reported that *G34* mutations exerted characteristic methylomic effects, regardless of the tumor tissue of origin, and this mutation could affect chromosome instability [97]. Although *G34R/V* mutations are relatively rare in pHGG, this genotype is likely to have specific methylomic signals and show extensive infiltration and various histological phenotypes [96,97].

### 5.2. Targeted Therapy for pHGG

Clinical trials of targeted therapy for pHGG are summarized in Table 2 and Table 3. *H3K27M* has been shown to inhibit PRC2, a multiprotein complex responsible for the methylation of *H3* at lysin 27, by binding to its catalytic subunit, EZH2 [98]. Mohammad et al. demonstrated that *H3K27M* mutated tumors require PRC2 for proliferation, and EZH2 inhibitors cease tumor cell growth [99]. EZH2 is a potential therapeutic target for the treatment of *H3K27M* mutated tumors, and the pediatric MATCH study included this targeted therapy. In Japan, a phase I clinical trial of a dual EZH1/2 inhibitor for pediatric, adolescent, and young adult patients with malignant solid tumors is in progress.

In the treatment of patients with DIPG, the Individualized Therapy For Relapsed Malignancies in Childhood (INFORM) registry study offered comprehensive molecular profiling of high-risk tumors to identify targetable alterations for precision therapy [100]. In this study, tumor material was obtained from brainstem biopsy, and molecular information was used for initiation of targeted therapy in 5 of 21 patients [100]. In addition to the INFORM study, the Pacific Pediatric Neuro-Oncology Consortium (PNOC003) study demonstrated the feasibility of genomic-based precision therapy for DIPG using NGS [101]. This study revealed that the molecular profiling of DIPGs changed the therapy and course of tumor progression [101]. More interventional molecular matching studies for DIPG are expected in the near future.

Some recent clinical trials of targeted therapy for pHGG have failed to prove its efficacy and safety. Sunitinib, a multi-targeted tyrosine kinase inhibitor that inhibits vascular endothelial growth factor receptor (*VEGFR*), platelet-derived growth factor receptor (*PDGFR*), and stem cell factor receptor (*KIT*), had no antitumor activity against HGG and ependymoma [73]. Although the combination of perifosine, an *AKT* inhibitor, and temsirolimus, an *mTOR* inhibitor, for CNS tumors, including HGG, was well tolerated, sufficient antitumor activity was not demonstrated [74]. The combination of dasatinib, *PDGFR-A* inhibitor, crizotinib, and *c-Met* inhibitor, for recurrent/progressive HGG or DIPG was poorly tolerated and its antitumor activity was minimal [75]. However, as the other targeted therapy for DIPG, phase II studies of ONC201, a dopamine receptor D2 antagonist for patients with newly diagnosed DIPG harboring the *H3K27M* mutation, demonstrated clinical efficacy [76]. Dopamine receptor D2 is a G protein-coupled receptor that promotes tumor growth [102]. In gliomas harboring the *H3K27M* mutation, dopamine receptor D2 is overexpressed and dopamine receptor D5 is suppressed, resulting in enhancement of sensitivity to a dopamine receptor D2 antagonist [103].

In addition, immune checkpoint inhibitors have been expected to bring a paradigm shift in treatment for gliomas. These have been shown effective in treatments for other malignancies, such as melanoma [104,105]. However, previous clinical trials failed to prove the outcome impact of immune checkpoint inhibitors in the treatment of adult GBMs [106,107]. On the other hand, microsatellite instability, which is used as a molecular marker for defective DNA mismatch repair genes, is detected more frequently in pediatric gliomas than in adult gliomas [108,109], suggesting that pediatric gliomas have a promising response to immune checkpoint inhibitors.

## 6. Infantile Gliomas

Compared to gliomas in children and adults, infantile gliomas have paradoxical clinical behavior. Whereas HGGs have a better clinical course [82,110], LGGs have a higher mortality rate [111,112,113]. Infantile pediatric gliomas have been reported to be subclassified into the following three molecular groups [114]:Group 1, hemispheric receptor tyrosine kinase-driven gliomas, including *ALK*, *ROS1*, *NTRK*, and *MET* fusions, which are enriched for high-grade glioma, and have an intermediate clinical outcome.Group 2, hemispheric RAS/MAPK-driven gliomas, which demonstrate excellent long-term survival with minimal post-surgery clinical intervention.Group 3, midline RAS/MAPK-driven gliomas, which are enriched for LGG, such as PA, with *BRAF* alternations, and have a poor outcome.

Because each subtype indicates the clinical and molecular features, updating the diagnosis and treatment of these gliomas is necessary. Additionally, because these molecular features can be targetable, novel molecular targeted therapies are expected to develop. The involvement of the RAS/MAPK pathway in infantile gliomas has been revealed, and current molecular profiling have identified novel alterations regarding this pathway.

### 6.1. Fusion Genes in Infantile Gliomas (Table 1)

*NTRK* fusions: *NTRK1*, *NTRK2*, and *NTRK3* are actionable drivers of tumor growth [115,116], and these genes encode the tropomyosin receptor kinase (TRK) family of receptor tyrosine kinases, proteins TRKA, TRKB, and TRKC, respectively. The TRK family plays a role in neuronal development, cell survival, and cellular proliferation [117]. *NTRK* fusions affect both the RAS/MAPK and PI3K/AKT/mTOR pathways, and are considered to be associated with tumorigenesis [118,119]. Wu et al. reported that 40% (4/10) of non-brainstem high-grade gliomas in children under 3 years of age contained *NTRK1-3* fusion genes [89]. *NTRK1-3* fusion genes have not been identified frequently in pLGG and adult GBM [33,40,120]. However, *NTRK* fusions can be detected in histologically diagnosed glioneuronal tumors [121], and cancer multi-gene panel tests should be widely and frequently performed.

*ALK* fusions: the *ALK* gene is thought to be associated with development and function of the nervous system. *ALK* fusion is reported to cause ectopic expression of the *ALK* fusion protein [23,122], resulting in upregulation of the RAS/MAPK and PI3K/AKT/mTOR pathways [114]. Although gliomas with *ALK* fusion are rare and the published literature is limited to several case studies, *CCDC88A-ALK* and *PPP1CB-ALK* were reported as the most frequent alterations [114,123,124,125]. In our institution, we encountered a case of HGG that harbored *VCL* as a novel partner of the *ALK* fusion gene [126].

*ROS1* fusions: ROS1 is an orphan tyrosine receptor, which is considered to be associated with cell proliferation and differentiation. Although *ROS1* fusion in glioma is quite rare, several reports have highlighted it as a targetable genetic alteration [127,128,129]. *GOPC-ROS1* was reported as the most common *ROS1* alteration in glioma. In addition, *CEP85L-ROS1*, *ZCCHC8-ROS1*, and *KLC1-ROS1* have also been identified [127,128,129]. The targeted agents for lung cancer with *ROS1* has shown significant antitumor activity [130,131].

### 6.2. Treatment for Each Group in Infantile Gliomas

Clinical trials of targeted therapy for infantile glioma are summarized in Table 2 and Table 3. As infantile gliomas are mostly single-driver tumors, they are suitable for precision therapy [114]. The efficacy of some types of targeted kinase inhibitors has already been demonstrated.

Group 1 tumors: group 1 tumors harbored *ALK/ROS1/NTRK/MET* alterations, and 5-year OS was 53.8, 25.0, and 42.9% for tumors with *ALK*, *ROS1*, and *NTRK* fusions, respectively [114]. Entrectinib, an oral inhibitor of the tyrosine kinases TRKA/B/C, ROS1 and ALK, was evaluated in two phase I studies, and shown to be well tolerated. Only reversible grade 1/2 adverse events and active against gene fusions of *NTRK1-3*, *ROS1*, and *ALK* were found in adult patients with solid tumors, including CNS tumors [132]. Response was observed as early as 4 weeks after administration, and lasted as long as >2 years [132]. A phase I clinical trial, using larotrectinib for newly diagnosed pHGG with *NTRK* fusion, is planned in the United States. *NTRK1-3*, *ROS1*, and *ALK* fusion genes in infantile glioma are targetable, therefore, nationwide adaptation of NGS to evaluate these fusion genes and more extensive accumulation of clinical data are required.

Group 2 tumors: because group 2 tumors show excellent long-term survival, a safe resection and careful follow-up are recommended [114].

Group 3 tumors: Because most group 3 tumors result in poor outcomes after conventional chemotherapy, targeted therapy, such as *BRAF*/*MEK* inhibitors, should be administered as soon after initial diagnosis as possible [114]. In the treatment of these tumors, dabrafenib, a *BRAF* inhibitor, and selumetinib, an *MEK* inhibitor, have been reported to be effective and tolerable [68,133,134].

## 7. Conclusions

Because the molecular characteristics of pediatric gliomas are different from those of adult gliomas, molecular genetic analysis is essential to diagnose and treat pediatric gliomas. Because targetable molecular aberrations can be detected in pediatric gliomas, searching for these aberrations is very important. Moreover, sufficient evidence of the novel targeted therapies has not been demonstrated in the present clinical trials. However, efficient targeted therapies are expected to be feasible even for rare subtypes in CNS tumors because of multi-gene panel analysis using NGS in the near future. Although the era of incorporating molecular genetic analysis into clinical practice is beginning, the search for specific molecular aberrations in pediatric gliomas is insufficient. How to administer molecular genetic analysis in clinical practice may be a future issue.

## Figures and Tables

**Figure 1 cancers-13-00758-f001:**
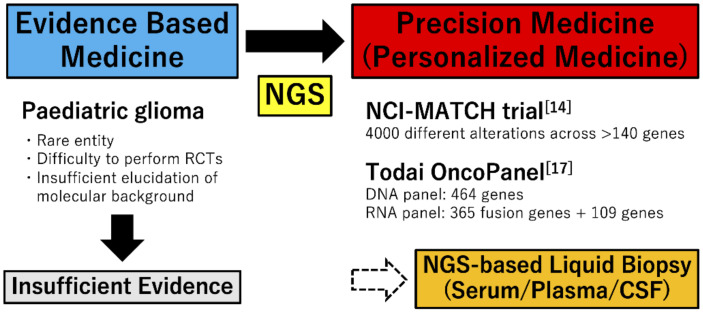
Change of treatment strategies for pediatric glioma. In treatment for pediatric glioma, because of advances of NGS, precision medicine is expected instead of evidence-based medicine. NCI-MATCH trial and Todai OncoPanel are representative precision medicine for pediatric glioma, and liquid biopsy is also expected in the future. RCT, randomized controlled trial; NGS, next-generation sequencing; CSF, cerebrospinal fluid.

**Table 1 cancers-13-00758-t001:** Summary of molecular landscape in pediatric gliomas.

Molecular Alteration	Function	Tumor Type	Potential Biomarker
pLGG (RAS/MAPK pathway)			
*KIAA1549-BRAF* fusion	Activation of *BRAF* kinase domain Deregulation of the RAS/MAPK pathway	PA	Diagnostic marker Poor prognostic marker
*BRAF* V600E mutation		PA/PXA/GG/DA	
*FGFR1*	Upregulation of the RAS/MAPK pathway	CNS tumors	NA
*NF-1*	Negative regulator of RAS	PA/DA	NA
pLGG (non-RAS/MAPK pathway)			
*MYB* and *MYBL1*	Control of proliferation and differentiation of hematopoietic and other progenitor cells	DA	NA
*CDKN2A* homozygous deletion	Non-coding of the gene for tumor suppressors, protein p14^ARF^ and p16^INK4A^	PXA	Poor prognostic marker
pHGG			
*H3K27M* mutation	Decrease levels of lysine 27 methylation	GBM/DIPG	Diagnostic marker Poor prognostic marker
*H3G34R/V* mutation	Changes the distribution of lysine 36 methylation		
Infantile glioma			
*NTRK* fusions	Upregulation of the RAS/MAPK and PI3K/AKT/mTOR pathways	Hemispheric HGG	Intermediate prognostic marker
*ALK* fusions			
*ROS1* fusions			

pLGG, pediatric low-grade glioma; MAPK, mitogen-activated protein kinase; PA, pilocytic astrocytoma; PXA, pleomorphic xanthoastrocytoma; GG, ganglioglioma; DA, diffuse astrocytoma; CNS, central nervous system; NA, not available; pHGG, pediatric high-grade glioma; GBM, glioblastoma; DIPG, diffuse intrinsic pontine glioma.

**Table 2 cancers-13-00758-t002:** Summary of recent clinical trials of targeted therapies for pediatric glioma.

Author	Year	ClinicalTrials.gov ID	Phase	Patients	Disease	Molecular Target	Treatment	Outcome
LGG								
Fangusaro et al. [68]	2019	NCT01089101	II	3–21 y *n* = 38	Recurrent/refractory LGG	*MEK*	Selumetinib	Positive antitumor activity Well-tolerated
Hargrave et al. [62]	2019	NCT01677741	I/II	2–18 y *n* = 32	Recurrent/refractory LGG	*BRAF* V600E mutant	Dabrafenib	Positive antitumor activity Well-tolerated
Nicolaides et al. [70]	2020	NCT01748149	I	3–17 y *n* = 19	Recurrent/refractory gliomas	*BRAF* V600E mutant	Vemurafenib	Positive antitumor activity Well-tolerated
HGG								
Wetmore et al. [73]	2016	NCT01462695	II	18 m–22 y *n* = 30	HGG or ependymoma	*VEGFR* *PDGFR* *KIT*	Sunitinib	No antitumor activity Well-tolerated
Becher et al. [74]	2017	NCT01049841	I	4–24 y *n* = 23	Recurrent/refractory brain tumor	*AKT* *mTOR*	Perifosine Temsirolimus	Well-tolerated
Broniscer et al. [75]	2018	NCT01644773	I	2–21 y *n* = 25	Recurrent/progressive HGG or DIPG	*PDGFRA* *c-Met*	Dasatinib Crizotinib	Minimal antitumor activity Poorly tolerated
Chi et al. [76]	2019	NCT03134131	II	3–42 y *n* = 18	*H3K27M* mutant diffuse midline glioma/DIPG	DRD2/3	ONC201	Positive antitumor activity

LGG, low-grade glioma; y, years; HGG, high-grade glioma; m, months; DIPG, diffuse intrinsic pontine glioma; DRD2/3, dopamine receptor type 2/3.

**Table 3 cancers-13-00758-t003:** Summary of ongoing clinical trials for targeted therapies in pediatric glioma.

ClinicalTrials.gov ID	Phase	Patients	Disease	Molecular Target	Treatment	Status
RAS/MAPK pathway targeted therapy						
NCT01734512	II	3–21 y	Recurrent/progressive LGG	*mTOR*	Everolimus	Active, not recruiting
NCT01748149	I	Up to 25 y	Recurrent/refractory glioma	*BRAF* V600E mutant	Vemurafenib	Active, not recruiting
NCT02684058	II	12 m–17 y	LGG or relapsed/refractory HGG	*BRAF* V600E mutant *MEK*	Dabrafenib Trametinib	Recruiting
NCT03363217	I/II	1 m–25 y	NF-1, Recurrent/refractory LGG	*MAPK/ERK* pathway *BRAF* fusion	Trametinib	Recruiting
NCT04485559	I	1–25 y	Recurrent grade 2 glioma	*MAPK/ERK* pathway *mTOR*	Trametinib Everolimus	Recruiting
NCT03429803	I	1–25 y	Recurrent/progressive LGG	*BRAF* fusion	TAK-580	Recruiting
NCT02285439	I/II	1–18 y	NF-1, Recurrent/refractory LGG	*MEK*	MEK162	Recruiting
NCT03696355	I	2–21 y	DIPC or other diffuse midline *H3K27M* mutant gliomas	*PI3K/Akt/mTOR*	GDC-0084	Active, not recruiting
NCT02650401	I/II	Up to 18 y	CNS tumor	*NTRK* or *ROS1* fusion	Entrectinib	Recruiting
NCT04655404	I	Up to 21 y	Newly diagnosed HGG	*NTRK* fusion	Larotrectinib	Not yet recruiting
Other targeted therapy						
NCT03749187	I	13–25 y	Gliomas, *IDH1/2* mutant	PARP	BGB-290 + TMZ	Recruiting
NCT03416530	I	2–18 y	Newly diagnosed DIPG Recurrent/refractory *H3K27M* gliomas	DRD2	ONC201	Recruiting
NCT01922076	I	37 m–21 y	Newly diagnosed DIPG	Tyrosine kinase WEE1	Adavosertib	Active, not recruiting

MAPK, mitogen-activated protein kinase; y, years; LGG, low-grade glioma; m, months; HGG, high-grade glioma; NF-1, neurofibromatosis type 1; DIPG, diffuse intrinsic pontine glioma; CNS, central nervous system; PARP, poly ADP ribose polymerase; TMZ, temozolomide; DRD2, dopamine receptor type 2.

## Data Availability

No new data were created or analyzed in this study. Data sharing is not applicable to this article.

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
