# Peer review of "Pediatric Glioma: An Update of Diagnosis, Biology, and Treatment"

_cancers, 2021, doi:10.3390/cancers13040758_

Round 1

Reviewer 1 Report

This review offers a comprehensive overview of molecular markers and the results of recent clinical trials. In its current form it's hard to see how the identification of more and more markers that are unique to even more rare subtypes offers any hope for those affected by gliomas. Especially when many of the drugs that target more common genetic alterations have failed to demonstrate efficacy in clinical trials.

It hints at the fact that hope for the future lies in precision medicine and it would be good to see more detail on how the authors perceive this working for rarer subtypes. I think it would also benefit from the addition of a figure depicting how future precision medicine trials could work and tie in to 'Onco-match', 'Todai-OncoPanel' and any other relevant drug finding software. What about efforts to identify new drugs and get them approved for use in children e.g. ACCELERATE. What samples should be analysed (tumour/serum/CSF), should the focus be on specific pathways or NGS?  How does this sit with the current preference for methylation profiling, small biopsies may prohibit both this and NGS profiling?

There is an opportunity here to produce a review that many will refer to as setting the scene for the way ahead rather than a rather dry summary. The second section is in my mind the most exciting part of this review and should be expanded.

Author Response

RESPONSES TO REVIEWER #1:

We thank Reviewer #1 for the insightful comments, which have helped to improve our manuscript.

Comment:

This review offers a comprehensive overview of molecular markers and the results of recent clinical trials. In its current form it's hard to see how the identification of more and more markers that are unique to even more rare subtypes offers any hope for those affected by gliomas. Especially when many of the drugs that target more common genetic alterations have failed to demonstrate efficacy in clinical trials.

Response:

We thank Reviewer #1 for this comment. We agreed that sufficient evidence of the treatment for pediatric gliomas has not been demonstrated in recent clinical trial. We explained this fact and added a prospect in the future in the Conclusion, as follows:

“Actually, sufficient evidence of the novel targeted therapies has not been demonstrated in the present clinical trials. However, efficient targeted therapies are expected to be feasible even for rare subtypes in CNS tumour because of multi-gene panel analysis using NGS in the near future.” (p. 11, lines 839–842)

Comment:

It hints at the fact that hope for the future lies in precision medicine and it would be good to see more detail on how the authors perceive this working for rarer subtypes. I think it would also benefit from the addition of a figure depicting how future precision medicine trials could work and tie in to 'Onco-match', 'Todai-OncoPanel' and any other relevant drug finding software. What about efforts to identify new drugs and get them approved for use in children e.g. ACCELERATE. What samples should be analysed (tumour/serum/CSF), should the focus be on specific pathways or NGS? How does this sit with the current preference for methylation profiling, small biopsies may prohibit both this and NGS profiling?

Response:

We thank Reviewer #1 for this comment. We agreed that additional information about precision medicine was necessary. As readers can understand relationship between precision medicine and drug findings software, we added Figure 1 in this article.

Besides, to introduce international organization for identification and approval of new pediatric oncology drugs, we added explanation about ACCELERATE, as follows:

“In addition to development of the molecular-based diagnostic technologies, international cooperation to identify and approve new paediatric oncology drugs, such as ACCEL-ERATE organized in Europe in 2015, has been advancing [18]. Many clinical trials for novel treatment are being conducted around the world. The era of incorporating mo-lecular genetic analysis into clinical practice is beginning (Figure 1).” (p. 3, lines 133–137)

Moreover, we also added explanation about NGS-based liquid biopsy and explained that sufficient tumor tissue samples obtained by conventional biopsy seems to essential for NGS or methylation profiling, as follows:

“Liquid biopsy is also one of the current topics as a less invasive method using body fluid, such as plasma or cerebrospinal fluid (CSF), for molecular diagnosis. Recent efforts in-cluding ours have developed digital PCR-based liquid biopsy targeting cell-free tumour DNA in CSF for detecting glioma-specific diagnostic mutations [15]. NGS-based liquid biopsy has been attempted. Miller et al. used NGS to analyze CSF samples from patients with diffuse gliomas, and identified glioma-related genetic alterations in 42 (49.2%) of 85 tumours [16]. Although they also detected these alterations from 3 (15.8%) of 19 plasma samples, all patients with positive plasma had radiological evidence for dissemination within the CNS [16]. These results indicated that the sensitivity of liquid biopsy is still an unsolved issue for the molecular diagnosis of CNS tumours. Accordingly, sufficient tissue samples obtained by conventional biopsy seems to be essential in the present state, es-pecially for NGS or methylation profiling.” (p. 3, lines 106–117)

Comment:

There is an opportunity here to produce a review that many will refer to as setting the scene for the way ahead rather than a rather dry summary. The second section is in my mind the most exciting part of this review and should be expanded.

Response:

We thank Reviewer #1 for this comment. We hope that the revised manuscript will be accepted for publication.

Reviewer 2 Report

In this manuscript, the authors summarized recent findings that provide an overview of paediatric gliomas. They also described recent developments in strategies for their diagnosis and treatment. This review brings together a wide body of information that may be of use for researchers interested in this field. My specific comment is as below.

Please summarize the molecular landscape in pLGG, pHGG, and infantile gliomas that mentioned in section 4.1, 5.1, and 6.1 as one or two tables. It will help readers to find this knowledge at a glance.

Author Response

RESPONSES TO REVIEWER #2:

We thank Reviewer #2 for the insightful comments, which have helped to improve our manuscript.

Comment:

In this manuscript, the authors summarized recent findings that provide an overview of paediatric gliomas. They also described recent developments in strategies for their diagnosis and treatment. This review brings together a wide body of information that may be of use for researchers interested in this field.

Response:

We thank Reviewer #2 for this comment. We hope that the revised manuscript will be accepted for publication.

Comment:

Please summarize the molecular landscape in pLGG, pHGG, and infantile gliomas that mentioned in section 4.1, 5.1, and 6.1 as one or two tables. It will help readers to find this knowledge at a glance.

Response:

We thank Reviewer #2 for this comment. We summarized the molecular landscape in pLGG, pHGG, and infantile gliomain Table 1.